# The *Dispositif* of Holography

**Jacques Desbiens**

Independent Researcher, Montreal, QC H2J 2L3, Canada; ijacqueshere@gmail.com

**Abstract:** The French word *dispositif*, applied to visual art, encompasses several components of an artwork, such as the apparatus itself as well as its display conditions and the viewers themselves. In this article, I examine the concept of *dispositif* in the context of holography and, in particular, synthetic holography (computer-generated holography). This analysis concentrates on the holographic space and its effects on time and colors. A few comparisons with the history of spatial representation allow us to state that the holographic *dispositif* breaks with the perspective tradition and opens a new field of artistic research and experimentation.

**Keywords:** *dispositif*; synthetic holography; analog holography; holographic space; time and colors; multiple perspective; kinetics; chromaticity

## 1. Introduction

> "One can never experience art through descriptions. Explanations and analyses can serve at best as intellectual preparation."
>
> (Moholy-Nagy 1928)

In 1970, Jean-Louis Baudry introduced the concept of *dispositif* in cinema theory (Baudry 1970). He analyzed the relationship between the spectator and the cinematographic representation through its technologies, settings, projection conditions, etc. Since then, this concept has spread in cinema theory and offers some interesting insights for those of us who are interested in the artistic potential of holography.

While the French word "*dispositif*" is usually translated in English as "apparatus", in French it has a broader meaning, especially in philosophy and art theory. A *dispositif* is a set of elements, devices, parameters and relations that constitute a scene, a place, a situation or an event. In any case, the *dispositif* is comprised of the apparatus and the display, as applied in given circumstances, but also the image production and presentation conditions, the content and even the viewer himself. A *dispositif* will determine the optical, the geometrical, as well as the cultural characteristics of a representation, a presentation and an observation of images. Whereas the basic holographic display may be as simple as a light source and the hologram itself, which is a high-resolution photographic emulsion, both placed at specific angles to each other, its *dispositif* also comprises the wall, room, ambient light, the specific optical characteristics of the hologram, and also the viewer. Holograms break with our tradition of graphic images made of pigments and marks on surfaces, or from light projected on screens or even pixels. Not only is holography a new media, it also has an approach to image making, a physical and optical foundation, a production process, display parameters and visual effects, which are completely different from any other imaging technology since the beginning of art. Consequently, the holographic *dispositif* presents an altered temporal and spatial coherence that can be manipulated by the artist. Perhaps initially unsettling to the viewer, the artist soon sees the possibility of new visual effects and narrative tools.

The purpose of this article is not to explain the technical characteristics of holography; it is easy to find books and resources on the web for that information. Rather, my intent is to present a few features

singular to the holographic *dispositif* that have a decisive impact on its aesthetic and artistic production and reception. My own experiments in synthetic holography (computer-generated holography) were aimed at analyzing these very features. As an art historian and artist, my work is based on a blend of historical research and artistic experimentations, testing observations and ideas from art history as well as my own, then elaborating concepts from this process. Thus, the end result of my holographic work is not simply artistic expression, but rather the developments of concepts and analysis.

The characteristics of holographic images that I explore can be applied to analog optical holography. However, some effects can be difficult to achieve without computer-generated content. The main difference between analog and synthetic holography is the source of the image. The first uses an optical setup to record the light wave's interferences from an object, while the second requires a similar optical setup to record the interference patterns of a set of computer-generated images, each image representing an angle of view on a scene. Consequently, synthetic holograms are composite images that show a partial view of space, but this process offers to the artist easier access to an array of visual effects by means of computer graphics programs.

Although further experimentation and analysis is greatly needed, there are three formal elements of the holographic *dispositif* that one must consider in both the creation and in the critique of art holography: the holographic space, time, and chromaticity.

## 2. Space

Hundreds of years of spatial representation using linear perspective led us to accept the cyclopean model in which everything is seen from a single point. The viewing hole in Filippo Brunelleschi's *tavoletta*, the single point of view used by Leon Battista Alberti (1404–1472) in his perspective method, and all the subsequent perspectivist developments are a demonstration of a dioptrical approach to the representation of three-dimensional space. Like projecting a scene through a lens, light passes through the focal point. Even in photography and its correlates, the cinematographic camera or the virtual camera in computer graphics software, the single point of view is imposed. It is an egocentric position, the center of the world. Perspective is a shackle. The holographic space breaks with this geometric convention.

In my synthetic hologram, *Tractatus Holographis* (2005), I presented a fictitious 16th century treatise on holography (Figure 1). Written in Middle French and containing animated 3D illustrations, a page turns when the viewer moves laterally. On the first and second page, the traditional perspective model is presented with its pyramidal field of view emanating from a single point. On the third page is a short explanation of the holographic setup and on the fourth page is an illustration of the holographic space as two joined truncated pyramids (for an analysis of the holographic space, see: Pepper 1989).

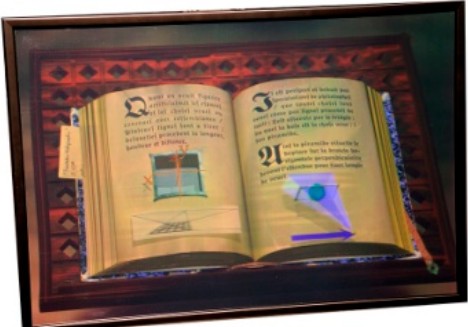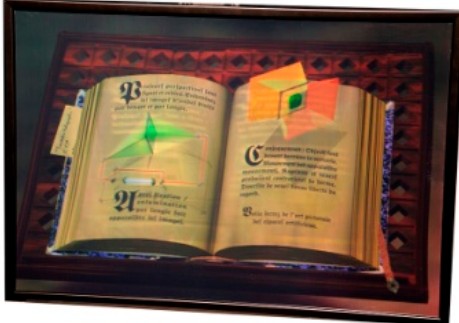

**Figure 1.** Tractatus Holographis, 2005. Two views of the synthetic hologram, 60 cm × 40 cm, created by author. (see: http://www.i-jacques.com/tractatusen.html).

The traditional perspective model places the viewer in the center of the world at a pivot point from which a single angle of view is projected on a plane. Instead, by reconstructing the light waves originating from a scene, holography creates a field in which the viewer can move freely and observe

the hologram content from many angles of view. The holographic *dispositif* transfers the pivot point to the hologram window and, therefore, expands the viewer's freedom of observation.

This is an important feature of the holographic *dispositif* that has significant consequences on its aesthetics and creative possibilities. The multiple perspectives of this continuous parallax optical system are what will usually define the holographic space as three-dimensional, or "3D". As an artistic representation, it means that what you see from here is different than what you see from there, that the frontal centered axis of view is only one angle among many others. Content can become variable with the viewer's movements without cinematic or mechanical animation.

In fact, this three-dimensionality affects the construction of a scene for holography in an unexpected manner. Voids occupy real optical three-dimensional spaces that determine the visibility of forms and the spatial relationship with the viewer. In the composition of a 3D holographic image, voids are not simply emptiness; they are volumes of space that are constructed by the artist to make "this" visible from "there".

Production techniques and the direction of illumination define whether the holograms are "transmission" or "reflection". Differentiating between these two types of holograms, Stephen A. Benton and V. Michael Bove, in their book *Holographic Imaging* (Benton and Bove 2008), present the analogy of holograms that are "windowlike" for transmission holograms, and "mirrorlike" for reflection holograms. Most holograms exhibited in art are "reflection holograms", which means that they are illuminated from the front. While these two categories refer to the orientation of illumination in the holographic process, this analogy can be extended to the holographic *dispositif.* Moving in front of a hologram, the viewer will see a variation of perspectives similar to the perspective variations that we see when we move in front of a mirror. Moreover, when we look at the history of the art of spatial representation, the mirror is sometimes referred to as an ideal. Alberti suggests using a mirror as a visual tool. The architect and sculptor Filarete (c.1400–c.1469), commenting on Brunelleschi's experiment on perspective, writes: "*It is certainly a subtle and beautiful thing to discover how to do it by rule from what the mirror shows you*". And for Leonardo Da Vinci (1452–1519), the mirror is "the painter's master". From a dioptric model, holography seems to reach into a catoptric model. While we can establish a comparison with images of the 2D world or sculptures of the 3D world, the holographic space is different because it is essentially an optical field in front and behind the hologram plane. When, in the 1990s, 3D computer graphics became accessible, it was often said that the artist had to think in 3D. In holography, the artist has to think spatially.

## 3. Time

The holographic *dispositif* establishes a distinctive spatial structure. However, the holographic space has an important impact on the representation of time that is more peculiar than any other imaging technology. In physics as well as in philosophy, space and time are interlinked. This is not different in holography and it is an important characteristic of my experiments. In my *Tractatus Holographis* hologram, a page seems to turn when the viewer moves laterally. In *Graphis* (2009) (Figure 2), texts in Arabic, Chinese, Latin, middle French, Greek and other content elements appear and disappear and the whole scene changes in synchronization with the viewer's movements.

In most of my synthetic holograms, there are more than a thousand computer-generated images with small variations that trigger content transformations in accordance to angles of view. This multiplicity of images allows us to introduce variations in the sequence that will result in kinetic effects when the viewers move in front of the hologram and see these variations. It is important to note that this is not cinematographic animation, but rather kinetics. Movements and transformations of content are synchronized and dependent on the viewer's movements. A form moving down, when the viewer walks toward the right, will appear moving up when the viewer moves left. Thus, holographic kinetics is reversible. Moreover, the speed of this moving form will be controlled by the viewer's speed, and if the viewer stops, the kinetics will stop too. In fact, we cannot even talk about speed or "x frames per second" in synthetic holography. Kinetics is a relationship between the content variations and the

viewer's movements. It is space + time. This creates the possibility of temporal distortions, such as "time-smear", a deformation or blur, resulting from harsh variations in the content (Figure 3). In most cases of time-smear, our two eyes do not see the same thing at the same time. This can twist, warp or contort the form. Depending on the configuration, the kinetic form may be stretched and apparently smudged. Temporal distortions are often seen as defects; however, this distortion may offer interesting artistic experimentations for artists.

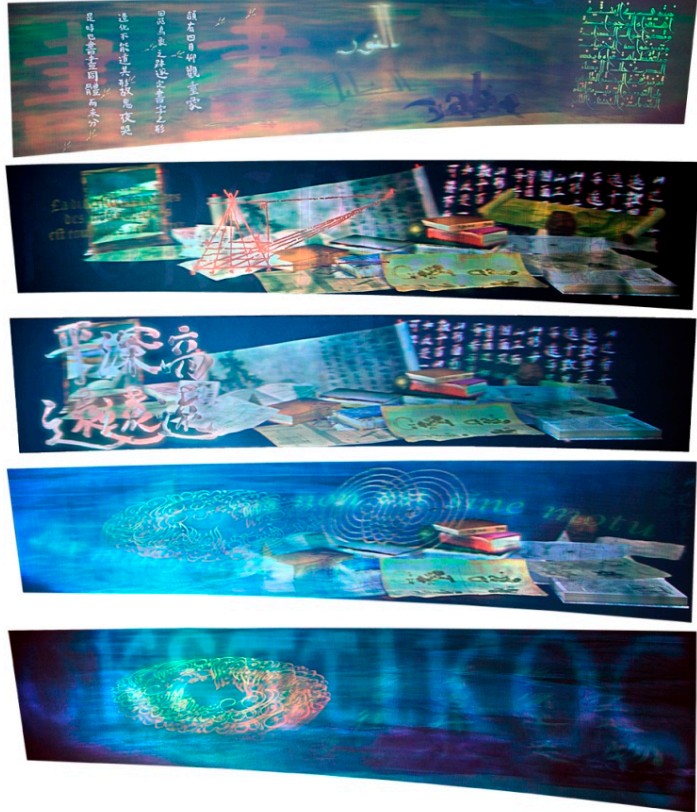

**Figure 2.** *Graphis*, 2009. Five views of the synthetic hologram, 3 m × 60 cm, created by author. (see: http://www.i-jacques.com/graphisen.html).

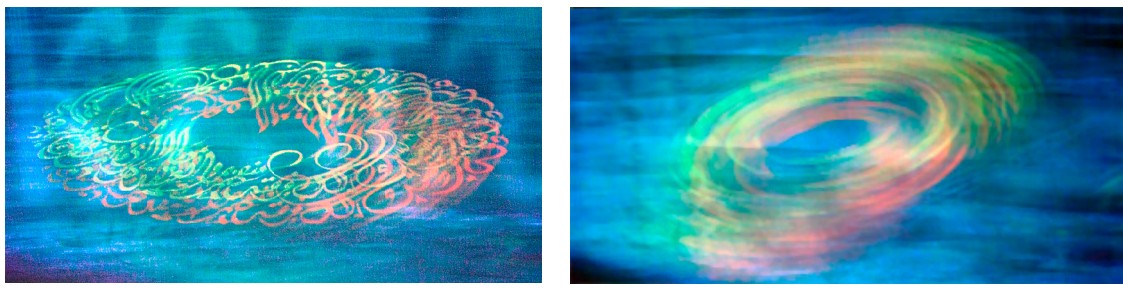

**Figure 3.** Time-smear of a rotation in *Graphis*, 2009.

Furthermore, multiple perspectives have an impact on the personal temporal relationship between the viewer and the content. While exhibiting synthetic holograms with some kinetics content, I noticed that the temporal relationship becomes an important part of the viewer's aesthetic experience. You may want to create a holographic artwork with a linear narration that can be viewed from left to right, but you do not control the viewer's movement. Contrary to cinema, people are free to move, look from the center, the left, moving from here to there . . . What you see from the right part of the holographic space can be different from what your friend will see from the left part . . . at the same time. The image

you see is your view and only yours. All other views are a different perspective. Thus, people often chose a particular angle of view, a personal space.

While the artist composes a space, time belongs to the viewer. Linear narration is shattered in holography. Visual information and its discursive attributes are contingent on spatial synchronization between elements and the viewers. Alignments, juxtapositions, superpositions, colors, transparent and reflective objects, directional vectors and focus points—all these formal elements and their significations can vary with multiple views and the spatial interactivity that holography provides. In a recent hologram, *Maze* (2018), I tried to use this spatiotemporal characteristic in the solution of a visual puzzle (Figure 4). The only way to solve this labyrinthine composition of stairs is for the viewer to move laterally and change viewing angles so that some parts of the stair's structure will appear or disappear. The movements of the viewers' bodies, the movements of their gazes, and the hologram's kinetics all contribute in this interactive hologram. "*Tempus non est sine motu*" (Time does not exist without motion) (Bacon 1267).

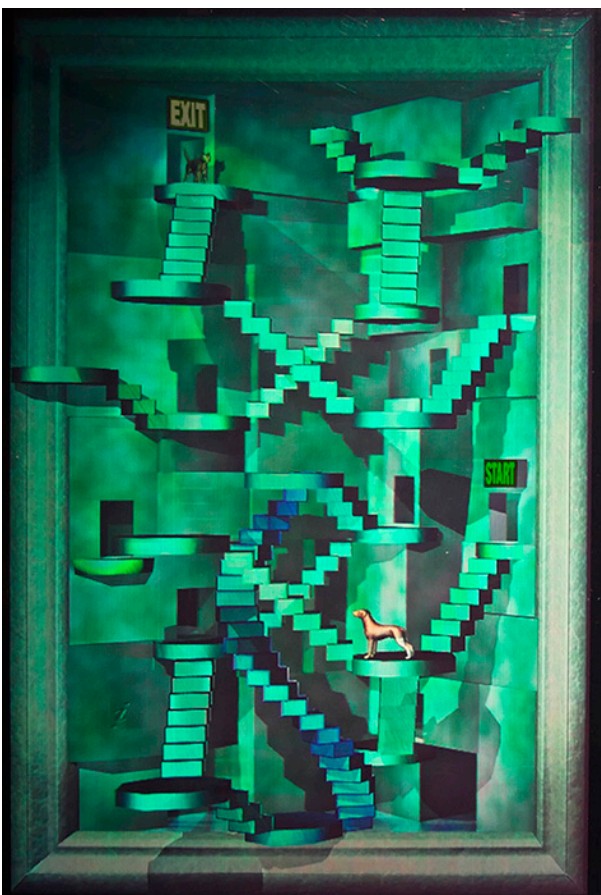

**Figure 4.** *Maze*, 2018. Synthetic hologram. 40 cm × 60 cm, created by author.

## 4. Chromaticity

We often forget that all our image displays have limited color ranges. Colors in paintings, photography, cinema, videos and computer screens have a smaller gamut than what is in nature's visible light spectrum. This is also the case in synthetic holography since the source images are computer generated; hence, the colors of a synthetic hologram derive from the standard RGB model used in these electronic displays. Still, the holographic *dispositif* will allow the artist to produce several chromatic effects synchronized with the viewer's movement. However, here analog holography seems advantageous when comparing the two processes. Red, green and blue lasers have a wider gamut than electronic displays and other color mediums. By using diffraction in the reconstruction of the

interference pattern, the holographic *dispositif* enters a chromatic field distinct from the traditional reflective or transmissive mediums. In many holographic artworks, these interactive light and color compositions are not achievable with any other medium.

Sometimes, these chromatic compositions are on the frontier between abstraction and figuration, as seen in the works of Rudie Berkhout or August Muth. They are true chromatic manipulations of the light spectrum. The holographic space tends to objectify content and allow for intentional ambiguity; color fields become volumes that display ethereal, optical and geometrical characteristics. Again, holography becomes a distinct medium that opens on a new form of "light painting", where chromaticity is manipulated in its very foundation.

## 5. Conclusions

Dependent on scientific developments and on access to expensive, complex and fragile equipment, artistic experimentations in holography are often difficult to grasp. Beyond all difficulties related to holography, accessibility is the key, not only for artists to produce holographic artworks, but also for people to see holograms in galleries and museums. Too often I have heard comments reducing holography to a technological curiosity or gadget. Obviously, this view is limitative and superficial. It ignores the singularity of the holographic *dispositif* and its impact on our relationship, physically and emotionally, with space, time and color. Furthermore, it obliterates the transformative nature of holography in the context of our history of artistic representation.

The artistic images that we create using various mediums, from painting to computer graphics, from photography to Virtual Reality, from cinema to its expanded forms, are all based on a long geometrical tradition. Holography is only a few decades old and distances itself from this tradition. Showing my holograms to people unaccustomed to the particularities of the holographic *dispositif*, I often had to tell them to move to appreciate the depth, the kinetic effects, the distortions, the variations. Viewers often approach a hologram the same way they do with any other 2D image displays, from a fixed single angle of view. In his text on *The cinema of attraction*, Tom Gunning identifies the early years of cinema as a time when the main objective was to show something in a different manner than theatre. Since its beginning, a lot of holograms have been "holograms of attraction", showing basic 3D or a peculiar visual effect. Much experimentation is still needed to develop a new visual language based on the attributes of the holographic *dispositif*, and artists must create new narrative forms that take advantage of its functions. Looking into the holographic art of the past decades, and the recent developments in technological and aesthetical possibilities that it offers, I think that we can apply to holography what Gunning states about cinema: "*Every change in film history implies a change in its address to the spectator, and each period constructs its spectator in a new way.*" (Gunning 2006).

Holography opens a window on an optical field breaking with centuries of imposed points of view and linear narrative forms. Moving from a dioptric model to a catoptric model, and then to a diffractive model, holography reaches into a new world of image making, where light, space and time are tangible materials.

**Funding:** This research paper didn't receive external funding.

**Acknowledgments:** I wish to thank Miss Mary Harman for her help in correcting the text and her support.

**Conflicts of Interest:** The authors declare no conflict of interest.

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
