# Peer review of "The Dispositif of Holography"

_arts, 2018_

Reviewer 1 Report

It may be worth introducing 'temporal and spatial coherence in the forward stage of the paper. The work is from practice based study, and we learn how the holographic space may be used to store images chronologically, providing an audience with a new experience with affective impact. This implies a 4-D dialogue previously unobtainable from the artists toolbox and provides digital holography/holographics a building lair for further scholarly study.

Author Response

Thank you.

Reviewer 2 Report

The article is organized in five clear sections: an introduction, the ‘three fundamental elements of the holographic dispositif’, and a conclusion. The sections that are better constructed are the last three. From line 106 on, the text is fairly clear, section three needs some syntax revision. Section two requires detailed revisions (mostly logical transitions and syntax) and the clarification of an odd sentence. The introduction, could offer a more direct definition of the dispositif in the first paragraph. As it is, the indirect definitions unfold lists of elements without a stable anchor. It does extend conceptual threads with the next following three sections and the conclusions, but these are relatively thin. When one reaches the end, the rest of the text makes full sense. In my experience, this type of issues - well written and structured sections that are accompanied by sections with problematic logical transitions and condense sentences - appear when the text  has previously been part of a larger research project or when parallel projects with similar objectives have been merged together. The introduction has traces of what seems that type of editorial cuts, there are also traces of literal translations from a language other than English; these elements make the text of the first three sections a little more difficult to follow. The last two sections do not show these issues. The objective stated at the beginning of the essay is sort of broad: to present a few features singular to the holographic dispositif and their impact on its aesthetic and artistic production and reception; the conclusions, however, offer a specific focus: the holographic dispositif is a key aspect of the construction of a new spectator. The article will benefit greatly by a few changes. The introduction and the second section could be better articulated with the conclusions, since they all give particular attention to the spectator’s dynamics.  This is an interesting and informative article that manages to make very good and clear points, in particular the need to think about dispositifs when talking about holography; the possibility of conceiving, identifying and describing a new type of spectator, and the potential outlining of the first key moments in the history of holography.  

Author Response

Thank you.

Reviewer 3 Report

The paper is interesting. I enjoyed the simplicity of its organization (space, time and colour). The experimentation is interesting and the developments seem sound. The paper has an excellent potential.

However, the paper still requires some work. The transitions are not smooth. Plus, some serious proofreading is needed.

When the author announces the organization of the paper, he/she does not really explain why  she /he chose that organization. He/she needs to justify it. The author should better guide the reader. He/she should lead the reader into his/her demonstration / reasoning.

Often, aside from proofreading, the writing lacks sharpness. For instance, it is quite clumsy to state  the french word means…but in french it has a broader meaning… . It is as if the author were opposing 2 french meanings, which is obviously not what he/she meant. The paper needs more clarity in the writing.

The author seems often lost with long lists of examples without explaining why each of them is relevant to his/her demonstration.  In other words, it is better to keep the list(s) short and fully explain why the example(s) the author chose is (are) relevant.

The conclusion is problematic  since it does not seem to refer much to the demonstration included in the corpus of the paper. I have noticed only a single sentence  ("It ignores the singularity of the holographic dispositif  and its impact on our relationship, physical and emotional, with space, time and colours.”) that directly refers to the corpus (and organization of the paper). The author should present here with more details the results of his demonstration/ reasoning and, only after, he/she may broaden either the subject or the conclusions he/she wishes to draw from the results. I did not have that the author has sufficiently elaborated on the results in this paper.

Author Response

Thank you.

Round  2

Reviewer 3 Report

The author has substantially improved his/her article. He has provided more detailed explanations about his/her process and reasoning. The editing has also helped. The author has better guided the reader in his/her demonstration. There are still some improvements to be made in terms of editing/proofreading (especially in terms of grammar and sentence structure), however, as a whole, the article is interesting, serious and well structured.